# Complete Genome Analysis and Animal Model Development of Fowl Adenovirus 8b

**DOI:** 10.3390/v14081826

**Published:** 2022-08-20

**Authors:** Aijing Liu, Yu Zhang, Jing Wang, Hongyu Cui, Xiaole Qi, Changjun Liu, Yanping Zhang, Kai Li, Li Gao, Xiaomei Wang, Yulong Gao, Qing Pan

**Affiliations:** 1State Key Laboratory of Veterinary Biotechnology, Harbin Veterinary Research Institute, Chinese Academy of Agricultural Sciences, Harbin 150069, China; 2Jiangsu Co-Innovation Center for the Prevention and Control of Important Animal Infectious Disease and Zoonoses, Yangzhou University, Yangzhou 225009, China

**Keywords:** FAdV-8b, complete genome, recombination analysis, pathogenicity, animal model

## Abstract

Inclusion body hepatitis (IBH), hydropericardium syndrome, and gizzard erosion associated with fowl adenovirus (FAdV) infection have caused notable economic losses worldwide. In 2020, severe IBH was observed in a layer chicken farm in Hebei Province, China. Liver samples were collected from layer chickens with severe IBH and virus isolation was performed in LMH cells. DNA sequence and bioinformatics analyses were conducted to determine the phylogenetic relationship and the pathogenicity assay was conducted in specific-pathogen-free (SPF) chickens. HeB20 strain was isolated and identified as FAdV-8b, and the complete genome was successfully sequenced (GenBank No. OK188966). Although widespread recombination in clinical strains has been reported within FAdVs, HeB20 showed some novel characteristics, and did not show any recombination, highlighting that recombinant and non-recombinant FAdV-8b coexist in the clinic poultry industry. Finally, pathogenicity animal model of HeB20 was developed and showed severe IBH and 10% mortality. Collectively, a new FAdV-8b strain (HeB20) was isolated and responsible for the severe IBH in layer chickens. Complete genome of HeB20 was sequenced and valuable for future epidemiological investigations. HeB20 was capable of inducing severe IBH and 10% mortality in SPF chickens; this animal model provides a powerful tool for the future vaccine development.

## 1. Introduction

Fowl adenovirus (FAdV) [1] belongs to the family Adenoviridea and genus Aviadenovirus, and is further divided into five species (FAdV-A-E) with 12 serotypes (FAdV-1-8a, 8b-11) according to the guidelines of the International Committee on Taxonomy of Viruses. FAdVs have been widespread throughout the world, such as in China [2,3,4], Pakistan [5,6], Chile [7], South Korea [8], Canada [9], Austria [7], Hungary [10], India [11], Japan [12], South Africa [13], Mexico [14], and Poland [15], causing substantial economic losses to the poultry farming industry.

Diseases such as inclusion body hepatitis (IBH), hydropericardium syndrome, and gizzard erosion are associated with FAdV infection, with IBH being mainly associated with FAdV-2, FAdV-11, FAdV-8a, and FAdV-8b infections [16,17,18,19,20,21,22]. IBH mainly occurs in broiler chickens (3 to 7 weeks of age), with 7 days and 20 weeks being the earliest and latest time points when IBH may appear, respectively [11]. Co-infection with other viruses and bacteria may further increase morbidity and mortality [17,23]. Recently, recombination between FAdV-E viruses has been reported [24], with possible constraints on species-specific genes and diversification patterns. For instance, recombinant FAdV AH720 strain harbors a genome backbone of FAdV-8b and the fiber gene from FAdV-8a, while recombinant GDLZ has a genome backbone FAdV-8a and the fiber gene from FAdV-8b [25,26]. Thus, recombination analysis of FAdVs may provide a powerful foundation for epidemiological investigations.

In the present study, a pathogenic FAdV-8b strain designated as HeB20 was isolated from chickens suffering from severe IBH in Hebei Province, China, and its whole genome was sequenced and analyzed. This HeB20 sequence demonstrated typical FAdV-8b characteristics compared to previously reported recombinant FAdV-8 strains (GDLZ and AH720). Furthermore, pathogenicity analysis showed that HeB20 could induce 10% mortality and severe IBH in SPF chickens, and this animal model would provide a powerful tool for vaccine evaluation in future studies. 

## 2. Materials and Methods

### 2.1. Viruses and Cells

FAdV strain HeB20 was isolated in Hebei Province, China, from clinical liver specimens of commercial layer chickens. The lethargic chickens showed ruffled feathers, severe IBH in livers, and a slight drop in egg production. Chicken Leghorn male hepatocellular (LMH, ATCC CRL-2117) cells were cultured in Dulbecco’s Modified Eagle’s Medium/nutrient mixture Ham F-12 (DMEM/F12) (Sigma-Aldrich, St. Louis, MO, USA) supplemented with 10% fetal bovine serum (Sigma-Aldrich, St. Louis, MO, USA), 100 IU/mL penicillin, and 100 μg/mL streptomycin, and incubated at 37 °C in a humidified 5% CO_2_ incubator.

### 2.2. Virus isolation, Purification, and Titration

LMH cell lines were used for the isolation of the clinical virus strain from this study. When the cell monolayer in a 75 cm^2^ flask reached approximately 80% confluence, the medium was aspirated and the cells were rinsed twice with PBS. Cell monolayers were infected with 5 mL each of 5-fold diluted liver homogenates and incubated at 37 °C for 2 h. The supernatant was aspirated and replaced with maintenance medium containing 2% FCS. The cells were incubated at 37 °C /5% CO_2_ and checked daily for cytopathic effects (CPE). The HeB20 was purified by three generations of plaque and titrated by 50% tissue culture infectious dose (TCID_50_).

### 2.3. Viral DNA Extraction and Sequencing

According to the manufacturer’s instructions, total DNA was extracted from the liver samples of chickens or HeB20 using a DNeasy Tissue Kit (Axygen, Union City, CA, USA). The DNA was used as a polymerase chain reaction (PCR) amplification template. The forward (F) and reverse (R) PCR primers FAdV-I F (5′-GCCACCGGAAGCTACTTTGA-3′) and FAdV-I R (5′-TTGTGATCCATGGGCATGA-3′) were designed based on the FAdV-I hexon gene and used to determine the FAdV genotype [27]. The whole genome of HeB20 was amplified with the primers in Table 1. The PCR products were sequenced by the Comate Biosciences Company (Jilin Comate Bioscience Co., Ltd., Changchun, China) and subsequently manually assembled using the Seqman program of the DNAstar software package (version 5.01, Madison, WI, USA) [28].

### 2.4. Transmission Electron Microscopy Examination

LMH cells infected with HeB20 strain at 0.01 multiplicity of infection (MOI) were harvested by freezing and thawing three times after incubating for 3 d at 37 °C in a 5% CO_2_ atmosphere. Culture media were centrifuged at 5000× *g* for 15 min to remove cellular debris. The supernatant was then centrifuged for 1 h at 15,000× *g*. The purified virus pellets were negative-stained and examined with a transmission electron microscopy examination (HITACHI H-7650).

### 2.5. Sequence Analysis

In addition to the experimental samples collected and prepared for sequencing, 18 FAdV isolates, including all 12 FAdV serotypes, were also selected for their complete genome sequence analysis. Then, a phylogenetic tree based on the complete genome sequence was constructed using MEGA 6.0 software (http://www.megasoftware.net/, accessed on 11 November 2020) by the maximum-likelihood method (1000 bootstrap replicates).

### 2.6. Experiment Animals

SPF chickens were obtained and maintained at the Experimental Animal Center of Harbin Veterinary Research Institute (HVRI, Harbin, China) of the Chinese Academy of Agricultural Sciences (CAAS, Beijing, China). The animal experiments were approved by the Animal Care and Use Committee of HVRI and performed following animal ethics guidelines and approved protocols.

### 2.7. Pathogenicity Experimentn

Ten 14-day-old SPF chickens were intramuscularly inoculated with 10^7.5^ TCID_50_ of HeB20 in 0.2 mL PBS. Ten SPF chickens served as negative controls and were inoculated with only PBS. The infection and control groups were monitored daily for 12 days. At 3 and 5 days post-infection (dpi), three chickens in the infection group and the control group were euthanized, respectively. The chickens’ liver, spleen, thymus, and bursa were collected from the infection and negative control groups 3 dpi, fixed, paraffin-embedded, and cut into sections. The tissue sections were stained with hematoxylin and eosin (HE) and microscopically examined to identify pathological changes caused by HeB20. Heart, liver, spleen, lung, kidneys, thymus, and bursa tissue specimens collected from the infection and control groups were also subjected to real-time PCR analysis to determine the virus distribution among the different tissue types.

### 2.8. Real-Time PCR

Real-time PCR was performed using a LightCycler 480 Real-Time Thermocycler (Roche, Rotkreuz, Switzerland). The primers were designed based on the L1 region of the hexon gene as follows: forward primer, 5′-GCCTACCCGCAATGTCACTA-3′; and reverse primer, 5′-ACCGAACCCGGTAACTGTTG-3′. The FAdV-8b TaqMan probe was 5′-(FAM)-AGCGGCTTCAGATCAGGTTC-(TAMRA)-3′. A 98-bp fragment containing the probe sequence was cloned into a pMD-18T vector, and 10^2^ to 10^9^ copies/µL were used as a PCR template to generate a standard curve. The chicken OVO gene was used as internal reference gene. The final concentration of PCR amplimers was calculated as copy numbers of FAdV-8b/OVO.

### 2.9. Statistical Analyses

Differences between the two groups were evaluated by Student’s *t*-test using GraphPad Prism software (GraphPad Software, La Jolla, CA, USA). Differences were considered statistically significant at *p*  <  0.01.

## 3. Results

### 3.1. Isolation and Identification of HeB20 Strain

A 1655-bp region of the FAdV hexon gene was PCR amplified (Figure 1A) and sequenced for the isolate obtained from the chickens with IBH. Then, the HeB20 was isolated in LMH cells, cytopathic effects (CPE) were observed after one passage (Figure 1B). The FAdV-8b virions were verified by electron microscopy. Negatively stained preparations showed nearly round particles with 70–90 nm (Figure 1C). Phylogenetic analyses of the partial hexon gene sequence confirmed that the pathogenic agent causing IBH in the chickens belonging to the genus Aviadenovirus, species E, serotype 8b, was designated strain HeB20 (Figure 1D).

### 3.2. Genome Organization of HeB20

Sanger dideoxy sequencing was performed, and the whole genome of HeB20 was determined as 43,843 bp in size with 58% G + C content, similar to that of other FAdVs. The open reading frames (ORFs) were predicted. A schematic of the position and relative size of all 40 ORFs that potentially encode functional proteins is shown in Figure 2A. The GenBank accession number of the complete genome nucleotide sequence of the HeB20 isolate reported in this study is OK188966. According to the complete genome sequence phylogenetic tree analysis, HeB20 belonged to the FAdV E8b (Figure 2B). Novel characteristics of HeB20 were identified in this study. For instance, the HeB20 repeat region was smaller than GDLZ and SD1356 but larger than other FAdV-E strains (Figure 2C).

### 3.3. Recombination Analysis of Fiber Gene

Fiber gene recombination was reported in recent isolates of FAdV-8a and 8b, AH720 contained a genome backbone originating from FAdV-8b and a fiber gene from FAdV-8a [26], GDLZ contained a genome backbone from originating FAdV-8a and a fiber gene from FAdV-8b [25] (Figure 3A). Interestingly, HeB20 had not recombined with a genome backbone and fiber gene originating from FAdV-8b (Figure 3A,B). The amino acid sequence similarity between the fiber of HeB20 and other FAdV-8b strains varied from 97.1% to 100%. Interestingly, the amino acid sequence identity of HeB20 fiber is 100% with the natural recombinant strain GDLZ (Figure 3C).

### 3.4. Pathogenicity of HeB20 in SPF Chickens

To evaluate the pathogenicity of HeB20 in chickens, 14-day-old SPF chickens were infected, and the clinical signs were monitored for 14 d. The clinical signs of the infected chickens were comparable. During the infection experiments, all chickens infected with HeB20 were lethargic and showed ruffled feathers. The clinical signs were associated with high morbidity (Figure 4A) but low mortality (Figure 4B).

Total DNA was extracted from the infected chickens’ tissues and detected by real-time PCR to evaluate the distribution of HeB20. High viral load was detected in the heart, liver, spleen, kidneys, thymus, and bursa at 3 and 5 dpi. The viral load in the liver was the highest among other tissues at 3 dpi. However, there was no significant difference among the tested tissues at 5 dpi (Figure 4C,D).

### 3.5. Histopathological Analysis

Histopathological analysis confirmed IBH in the infected liver specimens. Furthermore, substantial local infiltration of inflammatory cells leading to degeneration and necrosis of hepatocytes was observed, in addition to nuclear eosinophilic inclusion bodies. No significant damage was observed in the immune organs, including the spleen, thymus, and bursa (Figure 5).

## 4. Discussion

Since 2012, an increasing trend has been reported in the incidence of clinical cases of IBH in China, resulting in considerable economic losses to the poultry industry [29]. In 2020, severe IBH emerged in chicken farms of Hebei Province, China. In the current study, we successfully isolated a new serotype FAdV-8b strain (HeB20) from chicken livers with severe IBH. The determination of the phylogenic relationship of this new isolate was based on the serotype-specific sequence of the hexon gene. Furthermore, the complete genome of the pathogenic HeB20 was sequenced, and results were submitted to GenBank (accession number OK188966). 

Although the whole-genome sequence of HeB20 exhibited high similarity compared with that of other FAdV-8b strains, novel characteristics in the repeat region of HeB20 were also observed. These sequence results enrich the scientific knowledge regarding FAdV-8b and provide a reference for accurate and comprehensive diagnoses of IBH in chickens. Considering that the virus capsid of FAdVs plays a critical role in the virus life cycle [30], the major capsid (fiber) gene was analyzed in the present study. The sequence similarity between HeB20 and other FAdV-8b strains varied from 97.1% to 100%. The fiber protein can serve as an immunogen and target the immune response as the outermost capsid protein. Since vaccines based on the fiber-2 protein have already been developed [31,32,33,34,35,36], the sequence difference between fiber genes can provide a theoretical basis for developing vaccine components. 

Several vital details regarding FAdVs have been recently reported. For instance, widespread recombination has been observed in FAdVs; recombinant segments may be associated with major sites under positive selection, and mosaicism in genes may be targets of adaptive pressure toward an immune evasion strategy [24]. It has been reported that FAdV-8 strain AH720 has a genome backbone originating from FAdV-8b and a fiber gene originating from FAdV-8a [26]. Meanwhile, strain GDLZ consists of a 941-nt hexon gene, 1181-nt fiber gene, and 3101-nt ORF19 gene recombination region [25]. In the present study, recombination analysis indicated that HeB20 has not undergone recombination and is purely FAdV-8b. The recombination analysis of HeB20 provides a strong basis for formulating the strategies for the prevention and control of the IBH epidemic and a theoretical basis for the epidemiological investigation of FAdV-8a and FAdV-8b recombination.

Finally, we evaluated the pathogenicity of HeB20 in SPF chickens and observed high morbidity and low mortality. However, severe IBH was observed in swollen livers of infected chickens, confirming that HeB20 could induce severe IBH. The viral distribution among various tissues showed a higher viral load in the liver of chickens at 3 dpi, suggesting this may be the best tissue and time point for FAdV-8b isolation. Compared to the viral titers following infection with highly pathogenic FAdV-4, viral titers were significantly lower in the livers of SPF chickens infected with FAdV-8b, which may explain why FAdV-8b fails to cause high mortality [37]. Since we found that HeB20 caused severe IBH, the histopathological changes in immune organs were further evaluated in this study, indicating that HeB20 did not significantly affect the immune organs. The animal model presented in our study provides useful insights and powerful tools that will be beneficial for developing the FAdV-8b vaccine and understanding the deep pathogenesis of FAdV-8b.

## 5. Conclusions

We successfully isolated a new FAdV-8b strain (HeB20) responsible for severe IBH in Hebei Province, China. The complete genome of HeB20 was sequenced and analyzed, increasing our understanding of the molecular characteristics of FAdV-8b and enriching our knowledge regarding its diversity. Furthermore, recombination analysis provided theoretical support for future epidemiological investigations related to recombination between FAdV-8a and FAdV-8b. Moreover, the pathogenicity of HeB20 in SPF chickens showed that FAdV-8b could cause severe IBH and was one of the most important agents responsible for clinical IBH. Notably, the animal model developed in the current study provides a potent tool for vaccine development in the future.

## Figures and Tables

**Figure 1 viruses-14-01826-f001:**
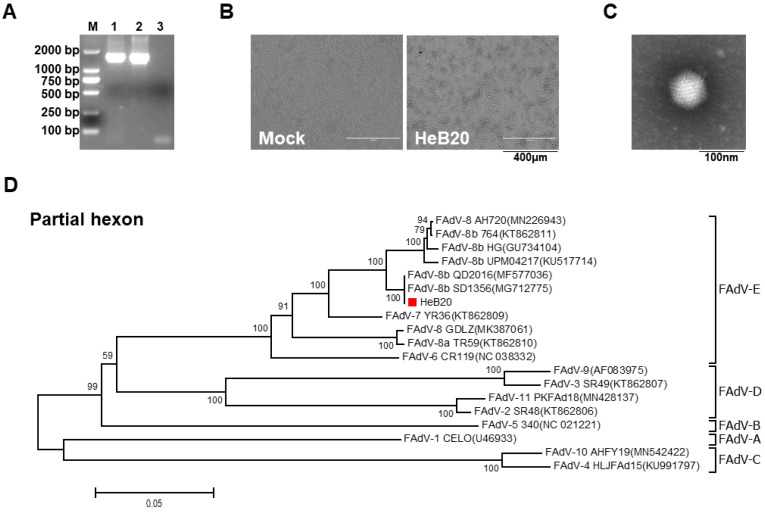
Characterization and purification of FAdV-8b strain HeB20. (**A**) PCR amplification of a 1655 bp region of the hexon gene. Line 1: PCR product of DNA template from clinical liver sample; Line 2: PCR product of DNA template from purified HeB20; Line 3: PCR product of DNA templated from mock LMH cells. (**B**) Typical cytopathic effect (CPE) of HeB20 infection in LMH cell cultures (arrow). Non-infected LMH cells showed no CPE. (**C**) Purified virus particles were obtained via CsCl_2_ gradient centrifugation. Scale bar = 100 nm. (**D**) Phylogenetic analysis of FAdV-8b strain HeB20 based on a sequence of the hexon gene. Accession numbers of the published sequence of the L1 hexon gene are provided in the parentheses.

**Figure 2 viruses-14-01826-f002:**
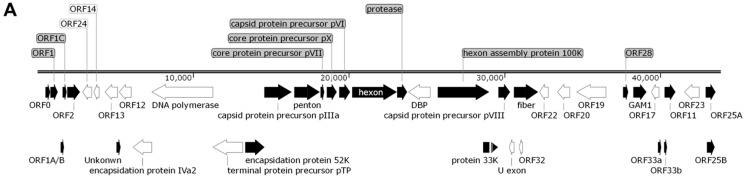
Schematic representation and phylogenetic analysis of HeB20. (**A**) Schematic representation of the HeB20 genome indicating the position of all 40 open reading frames. (**B**) Phylogenetic analysis of HeB20 based on whole-genome sequences. (**C**) Sequence analysis of the repeat region in FAdV-E.

**Figure 3 viruses-14-01826-f003:**
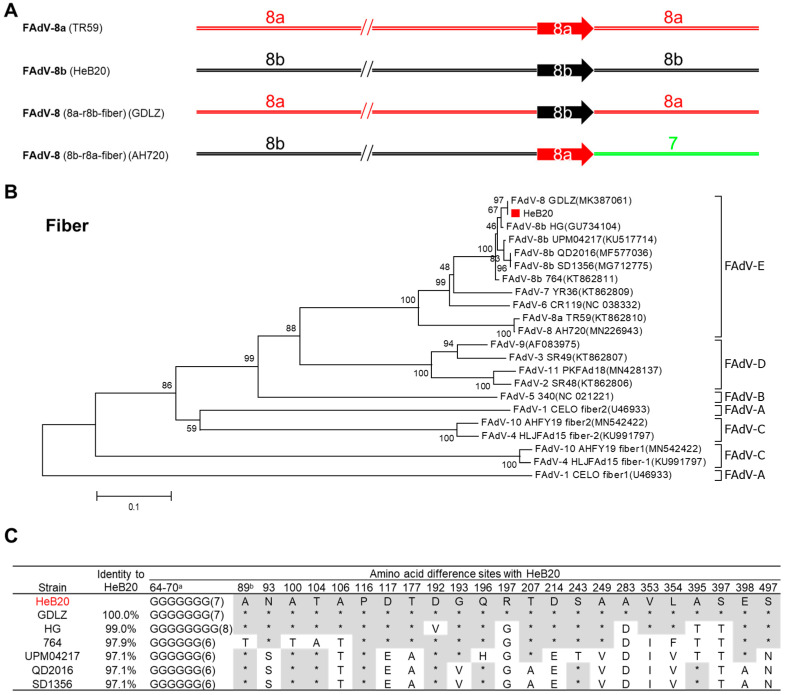
Recombinant and phylogenetic analysis of HeB20. (**A**) Recombinant representation of FAdV-8a (TR59), FAdV-8b (HeB20), FAdV-8 (8a-r8b-fiber; GDLZ), and FAdV-8 (8b-r8a-fiber; AH720). The arrow graphics represent fiber gene. The red, black, and green segments were from FAdV-8a, FAdV-8b, and FAdV-7, respectively. (**B**) Phylogenetic analysis of the major capsid (fiber) protein of FAdV-8b strain HeB20. Accession numbers of published fiber gene sequences are provided in the parentheses. (**C**) Sequence alignment analysis for fiber amino acid from different FAdV-8b isolates.

**Figure 4 viruses-14-01826-f004:**
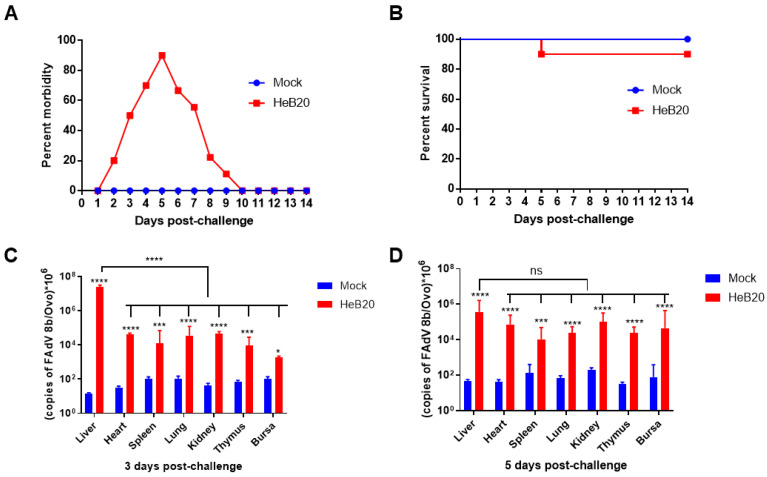
Pathogenicity analysis of HeB20. (**A**) Morbidity curves and (**B**) survival curves of chickens infected with HeB20. (**C**) Virus distribution in chickens at 3 dpi. Virus concentrations in the livers were significantly higher than those in the heart, spleen, kidneys, thymus, and bursa (*p*  <  0.01). (**D**) Virus distribution in chickens at 7 dpi. There was no significant difference in the viral load of the heart, liver, spleen, lung, kidneys, thymus, or bursa (*p*  >  0.05). The significance of differences was determined using two-way analysis of variance; ns, not significant, * *p* < 0. 05, *** *p* < 0.001, and **** *p* < 0.0001.

**Figure 5 viruses-14-01826-f005:**
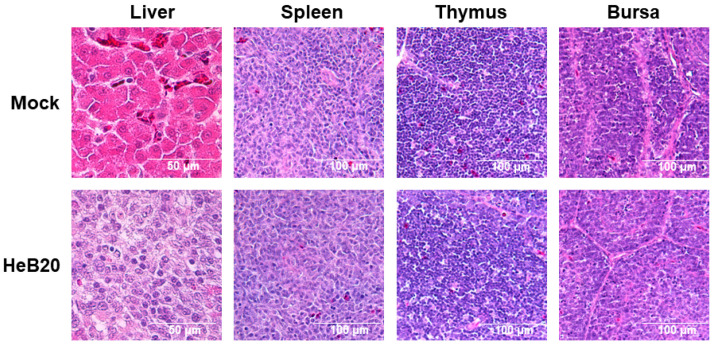
Histopathological analysis of chickens infected with HeB20. Degeneration and necrosis of hepatocytes and nuclear eosinophilic inclusion bodies showed in the liver. No significant histopathological changes were observed in immune organs (spleen, thymus, and bursa). Scale bar, 50 μm.

**Table 1 viruses-14-01826-t001:** The primers for complete genome sequencing of HeB20.

Primer	Sequence (5′-3′)	Primer	Sequence (5′-3′)
FAdV-8b 1F	CATCATCTATATATATCTAC	FAdV-8b 18F	CATGCCCATGGACCACAACAC
FAdV-8b 1R	CAGCCGGAATCGACAGACAT	FAdV-8b 18R	AGTACATGCGCTCCTGGTTGC
FAdV-8b 2F	GGACGGAGTCGATTTGGTAC	FAdV-8b 19F	CGACAGGTGTTTCGAGCTGG
FAdV-8b 2R	GTGTTGCGTCTCGACTGACG	FAdV-8b 19R	CGAGAAGCCTCTGAACGATCC
FAdV-8b 3F	CAGAATACGGTTACTTTACG	FAdV-8b 20F	CATGCCGTAGACCATGGCG
FAdV-8b 3R	CGCATCTCGCGCTGGATC	FAdV-8b 20R	CCGTCCGAGTCGATGTACAC
FAdV-8b 4F	GTTGAGACAGGTGCAGTC	FAdV-8b 21F	ACGACGGCGAGAACAATGG
FAdV-8b 4R	CTACGATGCCGCGCTCTTC	FAdV-8b 21R	GTTGTAGGTGACGCCGTGG
FAdV-8b 5F	CGGTTGCGTTTCGACGAAGG	FAdV-8b 22F	GGAACTCTACAAGGCGATGC
FAdV-8b 5R	GCGCAGCTAGTCAAGCGC	FAdV-8b 22R	GGTACTCAGATCCTCCTCGG
FAdV-8b 6F	GTGCCGAGCGAACTGCTCG	FAdV-8b 23F	GACTCCCAGAGCGACTACG
FAdV-8b 6R	GGAAGGATCTCACGCAGTAC	FAdV-8b 23R	CCTGGGTGCGGAGGAAGTAC
FAdV-8b 7F	GATCTCGTCACAGTCCTCG	FAdV-8b 24F	CTCACCGAAGGTCGAGTGC
FAdV-8b 7R	TCCACGACGACATGAACATC	FAdV-8b 24R	GGCGAATGCATTCCATTGTGTG
FAdV-8b 8F	GATGCCTTGTTTGTCGTCCT	FAdV-8b 25F	GGCTCACACTCAACTATGAC
FAdV-8b 8R	ACGACTTCTACAACGTGCAC	FAdV-8b 25R	CAATTGGACTGTGAGCAGC
FAdV-8b 9F	CAGCGATGGAGCCTCGTG	FAdV-8b 26F	CATAACCTCTGAGCGGCCATGATC
FAdV-8b 9R	GAAGGCATGGACGAGGCAG	FAdV-8b 26R	CGAATGCGGAGCCATATGG
FAdV-8b 10F	GTCGTCGTTGGTGTTGATGGC	FAdV-8b 27F	GCACGGTCCCGTCTTGAG
FAdV-8b 10R	GACCTGCCTCACCGATTGC	FAdV-8b 27R	CTTGCAGAAGAGGTGTTGTG
FAdV-8b 11F	CGCTCGTAGGAGGATGTCG	FAdV-8b 28F	GTTACCATGCAGTAGTGCGCAG
FAdV-8b 11R	GCTGAGCCAGCATGGTGTTC	FAdV-8b 28R	GTAGATCGTCTGTGTTCTTCTG
FAdV-8b 12F	GATGCGCAACAGTTGCCAC	FAdV-8b 29F	GTGCAATCGATGGTAGACAC
FAdV-8b 12R	CGTCGGACGGAATCCTATCC	FAdV-8b 29R	GGCAGAATCACAGCATTGAG
FAdV-8b 13F	CCTGCAGTTCAACGAGTACA	FAdV-8b 30F	GTGATAGGCGGAGCTCTCTC
FAdV-8b 13R	CAGGATCGGTTGTCCAGTTG	FAdV-8b 30R	GTTCAACCTGCGGGACAGATAC
FAdV-8b 14F	GCTGAGCGACATCGACACG	FAdV-8b 31F	GTACAACGTGTCCGCGTATC
FAdV-8b 14R	ACGTCTCCTTCTTCGGCTG	FAdV-8b 31R	CTAAGATGGCCAGGAACACCGTAG
FAdV-8b 15F	TCTTCGTCGCCGACCGTTG	FAdV-8b 32F	CCTGGTACACTGACACCTTAG
FAdV-8b 15R	GAGCTTGCAGGGCCTGAATG	FAdV-8b 32R	GGAGGTGACTCTGACTACG
FAdV-8b 16F	CCTTCGGTCAGATCAAGCAG	FAdV-8b 33F	GATTGTGATAGCCAGCACCCG
FAdV-8b 16R	CGGTATCGTGGTACAGGAGG	FAdV-8b 33R	TATCTACTTAAAATACACTCC
FAdV-8b 17F	CTCGCGAAGCCTTCTTTAAC	FAdV-8b 34F	CAGTTACTGCTCCTTCTATGC
FAdV-8b 17R	GTGAAGGACCATCCTCTCATTC	FAdV-8b 34R	CATCATCTATATATATCTAC

## Data Availability

Sequence data generated in the present study that support the findings of this study are publicly available and were deposited in GenBank/NCBI/NLM under accession numbers OK188966.

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
