# Peer review of "Complete Genome Analysis and Animal Model Development of Fowl Adenovirus 8b"

_viruses, 2022, doi:10.3390/v14081826_

Round 1

Reviewer 1 Report

The MS was already reviewed by this Reviewer twice, but the authors still neglected some of the questions or comments, unfortunately. Please find a commented pdf file attached again.

Author Response

Manuscript ID: viruses-1858943

August 18, 2022

Dear editor, 

Thank you for your kind review of our manuscript. Those comments are all valuable and very helpful for revising and improving our paper, as well as the important guiding significance to our research. We revised the manuscript in accordance with the reviewers’ comments, and carefully proof-read the manuscript to minimize typographical, grammatical, and bibliographical errors. The resubmitted manuscript is a novel version according to the reviewers’ comments and the details are listed in the attachment.

Reviewer 2 Report

Avian adenovirus is prevalent nowadays. In this MS, a lot of work by the authors on the isolation, genome analysis and pathogenicity of a novel FAdV-8b strain is warranted to improve our control strategy of FAdV-8b. Generally, the work is interesting, and here I have several minor suggestions to improve the manuscript.

1.In Materials and Methods 2.1 part, can authors offer Leghorn male hepatocellular (LMH) cells source information?

2.     Figure 4: What does *** indicate in Figure 4C?

3. In abstract part, authors displayed that HeB20 was capable to induce severe IBH and 10% mortality in SPF chickens and provides a powerful tool for the future vaccine development. However, authors did not discuss on adenovirus virulence standard in the manuscript, which maybe important information for readers.

Author Response

Manuscript ID: viruses-1858943

August 18, 2022

Dear editor, 

Thank you for your kind review of our manuscript. Those comments are all valuable and very helpful for revising and improving our paper, as well as the important guiding significance to our research. We revised the manuscript in accordance with the reviewers’ comments, and carefully proof-read the manuscript to minimize typographical, grammatical, and bibliographical errors. The resubmitted manuscript is a novel version according to the reviewers’ comments and the details are listed in the attachment.

This manuscript is a resubmission of an earlier submission. The following is a list of the peer review reports and author responses from that submission.

Round 1

Reviewer 1 Report

The present manuscript describes isolation and characterization of a fowl adenovirus originating from layers with IBH in China.

In my opinion this manuscript should not be published in its current form. There are several problems which need to be adressed .

  • From an epidemiological / clinical viewpoint, it contains only insufficient information concerning the origin of the virus and the case history (e.g. morbidity/mortality, clinical signs, production data, and pathology in layers). Consequently, it is impossible to estimate the production impact of this 'new' virus.
  • Already FAdV-8b strains have been described associated with IBH in chickens and have been investigated accordingly (also in China: Changjing et al 2016, Huang et al 2018, Chen et al 2019). Overall, the investigated strain HeB20 is described as a typical FAdV-8b without recombination. How will the results contribute to support national control strategies? Though novel characteristics of HeB20 are hinted at (L144-146) they are not described in detail. Furthermore, the described methodology fails to mention - among other thing – exclusion of possible co-infections, virus purification technique, virus titration and primers used for sequencing.
  • Animal infection models using FAdV-8b have been established previously (Steer et al 2015, Oliver-Ferrando et al 2017, Matos et al 2018, DeLuca et al 2020, Sabarudin et al 2021..) and the present model does not offer actual new insights. Furthermore, the terminology of clinical signs is not clearly understandable (e.g.: 'sleepy curliness') and gross findings from the animal trial are omitted entirely. Finally, the depiction of viral load in organs of mock-infected SPF (Figure 4 C+D) leads to doubt about the procedure.

Also - use of outdated/ wrong  virus taxonomy ('Fowl adenovirus group I').

Reviewer 2 Report

The MS by Liu et al discusses relatively interesting findings, but several flaws need to be corrected. The MS was already reviewed by this Reviewer but unfortunately, the required edits were ignored by the authors almost completely. Pls see the attached PDF file with several recommended edits and questions.
